# Burns in Nepal: a participatory, community survey of burn cases and knowledge, attitudes and practices to burn care and prevention in three rural municipalities

Kamal Phuyal,[1] Edna Adhiambo Ogada [ID],[2] Richard Bendell,[3] Patricia E Price [ID],[2,3] Tom Potokar[2,3]

[1]SAGUN: A Search for Harmony, Kupandol, Nepal
[2]Centre for Global Burn Injury Policy and Research, College of Human and Health Sciences, Swansea University, Swansea, UK
[3]Interburns, Swansea, UK

**Correspondence to**
Professor Patricia E Price;
p.e.price@swansea.ac.uk

## ABSTRACT

**Objectives** As part of an ongoing, long-term project to co-create burn prevention strategies in Nepal, we collected baseline data to share and discuss with the local community, use as a basis for a co-created prevention strategy and then monitor changes over time. This paper reports on the method and outcomes of the baseline survey and demonstrates how the data are presented back to the community.

**Design** A community-based survey.

**Setting** Community based in three rural municipalities in Nepal.

**Participants** 1305 households were approached: the head of 1279 households participated, giving a response rate of 98%. In 90.3% of cases, the head of the household was male.

**Results** We found that 2.7% (CI 1.8 to 3.7) of 1279 households, from three representative municipalities, reported at least one serious burn in the previous 12 months: a serious burn was defined as one requiring medical attention and/or inability to work or do normal activities for 24 hours. While only 4 paediatric and 10 adult cases in the previous 12 months reached hospital care, the impact on the lives of those involved was profound. Only one patient was referred on from primary to secondary/tertiary care; the average length of hospital stay for those presenting directly to secondary/tertiary care was 21 days. A range of first-aid behaviours were used, many of which are appropriate for the local context while a few may be potentially harmful (eg, the use of dung).

**Conclusion** The participatory approach used in this study ensured a high response rate. We have demonstrated that infographics can link the pathway for each of the cases observed from initial incident to final location of care.

## INTRODUCTION

The WHO has repeatedly tried to raise awareness of the fact that the majority of the burden of burn injury, from a global perspective, is borne by low and middle-income countries (LMICs).[1 2] Other recent publications[3] have confirmed that the greater burden

## Strengths and limitations of this study

► Detailed and sustained work with the local community to ensure enhanced community participation in the survey.
► Tracking individual patient journeys from injury to final destination for burn care.
► The use of infographics to overcome language and knowledge barriers when presenting the findings back to the community.
► Only one person with a burn injury per household was included in this study.

of mortality and morbidity related to burn injury is borne by the world's poor, residing mostly in LMICs. Patients in these countries continue to endure an extremely high level of suffering, with the rate of child deaths from burns being over seven times higher in lower resource settings than in the western environment.[4 5] Mongolia has a particularly high rate of child burn injury as identified by The Global Burden of Disease study in 2013[5]; detailed community studies, conducted in Ulaanbaatar, have supported this finding and shown that 27% (n=247/900) had a history of burn injury.[6] A recent systematic review[7] found eight epidemiological studies based in Nepal and concluded that burn injuries are the leading cause of morbidity and mortality in the country. The study authors report mortality rates from 4.5% to 23.5% and conclude that the limited data explain the lack of effective intervention programmes in the area.

The Annual Report from the Department of Health and Population for Nepal, covering the fiscal year 2016–2017, recorded 55090 burn or scald injuries nationally.[8] Population statistics for the same time period were 28.9

BMJ

million, giving an incidence rate of 0.19% of patients with an injury sufficiently severe to seek help from an outpatient facility where data were recorded. Data from public hospitals (92% response rate) and private facilities (47.2% response rate) were submitted as part of the Annual Report[8] and using the International Classification of Diseases codes T200–T301 for inpatients, 1710 cases were reported with 1006 related to those below the age of 19 and 586 under the age of 5 (81.7% of them aged 1–4). In the same time period, 46 in-hospital deaths were reported (31 women, 15 men) - although this does not include deaths from catastrophic events or from those who were sent straight to the mortuary.

Understanding the true morbidity and mortality impacts of burn injury in Nepal is challenging. Gupta *et al*[9] conducted a household survey using a cluster randomised cross-sectional countrywide survey in 15 districts in Nepal and reported a higher incidence of 55 burns from 1350 households (2%, 95% CI 1.5 to 2.6). The majority of cases (60.4%) were due to hot liquid and/or hot objects. The study authors conclude that the demographics and aetiology of burns at a population level vary significantly from hospital level data. Tripathee and Basnet,[7] in their systematic review, stress the importance of epidemiological data as a pre-requisite for planning and implementing prevention programmes at the community level.

To proactively develop lasting changes in resource-poor countries, it is vital that we have a detailed understanding of the current, local context including epidemiology, the availability of educational/prevention programmes and the ability of people in rural/remote areas to access vital first-aid treatment in order to plan appropriate interventions. This paper focuses on the first part of a major initiative to address these questions and bring about change using an Implementation Science approach based in Nepal. While previous studies start to give us an indication of the scale of the problem, they do not give us information about the ways in which people sought help, what they did immediately after the injury, what local knowledge exists to provide first aid and what prevention programmes are available locally. In order to develop a baseline for future comparison and ensure that we could fully evaluate our interventions, we needed to conduct a more in-depth community-based survey so that we could develop and co-create (with community stakeholders) a community prevention and first-aid strategy. This paper presents the findings of the baseline phase of the study. The specific objectives, related to selected areas of Nepal, were to (1) record the incidence and causes of burn injury in 2017–2018 and (2) explore the knowledge, attitudes and practices of the community regarding burn care and prevention.

## METHODS

The ongoing study uses a participatory action research design to develop a prevention and first-aid strategy at the rural municipality (RM) level (n=460 RM in Nepal).

This first phase included a community sensitisation and preparation plan, followed by baseline data collection. As the prevention strategies need to be developed at the RM level, baseline data was collected in one randomly selected municipality in each of three districts: Ama Chhodingmo RM in Rasuwa, Indrasarowar RM in Makawanpur and Laxminiya RM in Dhanusha. These districts were selected from three traditionally noted geographic regions of Nepal: Terai (flatland), Hills and Himalayan regions. This purposive selection was used to ensure a degree of diversity in the communities studied, while representing the geopolitical areas that any national policy would need to address.

The study population was all the people living in those rural municipalities, regardless of socio-demographic factors. A sample of 1305 households was selected from a population of 47 642, based on the WHO formula,[10] adjusted to account for the relative population size in each district. Household selection was completed using a systematic sampling approach drawing from a listing of the total number of households in each RM and formulated during the sensitisation preparation phase at each municipality office, when detailed discussions took place with local stakeholders.

Rasuwa district is located north of Kathmandu. It is a district of natural resources, particularly Langtang and Gosaikunda regions which are well known for their natural beauty and trekking routes with a high potential for ecotourism businesses. However, poor roads, poor access to university education and poverty have hindered the development of the district. Ama Chhodingmo is located in the mountain region, bordering China to the north, and it contains both natural and cultural diversity. The landscape in this area makes road construction very difficult, and most of the northern areas lack access to vehicle-friendly roads. Rasuwa itself is accessible by bus from Kathmandu via the Pasang Lhamu highway, but the administrative headquarters (Dhunche) are about 120 km from the capital. Over 98% of the population are Hill Ethnic Groups or Tamang; this is one of the indigenous inhabitant groups of Nepal, with their own distinct language, living in scattered hill villages.

Makawanpur is a hilly district, located on the southern border of Kathmandu. Despite bordering Kathmandu valley, many areas of Makwanpur remain inaccessible by road. The main livelihood for the majority of the population is subsistence farming. The district includes within its borders various agro-climatic zones: valleys, plains and mid-to-high hills. High rates of unemployment have driven a surge of migration - particularly of younger people - to Kathmandu and abroad. Indrasarowar RM, selected for this project, is located in the north-eastern part of the district in which the majority of the population belong to Tamang ethnic group (80%). Just over 9% of the population belong to the Newar ethnic group who are the historical inhabitants of the Kathmandu Valley, while a further 9.6% of the population are Hill Brahmin/Chhetri. Over 70% of the population speak Tamang and 26.5% speak Nepali.

Dhanusha is one of the southern Terai districts of Janakpur Zone and it is an agriculture-dependent district. Janakpurdham city is the capital of state 2 where this district is located and it is one of the commercial centres in the Terai region. It is also the administrative hub of the state. Laxminiya RM is located around 5 km from the major city of Janakpur; the settlement straddles a highway, allowing for some road access. The main livelihood is subsistence farming. The majority of people in this RM belong to Madhesi middle caste (ie, Yadav, Sudhi, Lohar, Sonar; this is a new term recently accepted into official vocabulary to describe this separate group in the Nepali social classification system) and Maithili is their main language. This is the most densely populated area included in this study.

Initial discussions with the key stakeholders included 15 Key Informant Interviews and 19 Focus Group Discussions with Female Community Health Volunteers, teachers, social workers and healthcare staff. This important first stage of this study, which took place over a 6-month period in the first half of 2018, allowed the researchers to develop their understanding of the general context of the burn injuries. Participant observation sessions (n=13) that focused on food storage and cooking practices, together with social maps and seasonal calendars, were used to augment the narrative on burn injuries, their prevention and management in each of the rural municipalities. Based on this initial work, items for the survey were prepared both in Nepali and English and reviewed by five local experts (including Sagun board members and university staff) to enhance their validity. The data collection tool asked households to report on any burn injury that required medical attention (which could include a medical post, a health worker, a Female Community Health Volunteer or a pharmacy) in the previous 12 months.

Survey data was collected in July and August 2018 using a structured questionnaire to collect data on socio-demographic and household descriptors, incidences of burn injury meeting the case definition, first aid received and health-seeking behaviour after the burn injury. The questionnaire was pretested in Mangaltar of Kavre district, where it was piloted on 15 households; only minor changes were required at the end of this phase. The final version of the survey tool was developed into an electronic format (CSPro software) in order to assist with the logistics of collecting data in remote areas, where smartphones are used routinely despite the geographic remoteness.

All enumerators, from a pool of experienced participatory research data collectors regularly used by Sagun, were trained in data collection using an electronic version of the form on smartphones to ensure consistency of use and reduce human error. The training included a day on theory and a day on practical issues, followed up by three supervised community-based training opportunities resulting in 3–5 full days of supervised training. All enumerators were instructed to call at the selected households up to three times until someone was present to complete the questionnaire: after three attempts no further attempt was made to contact the household members.

Three introductory visits were organised in each RM to discuss the details of the research proposal with representatives of the local government, healthcare staff and other relevant stakeholders at the RM level. Survey data was recorded and analysed using IBM SPSS V.23. The unit of analysis was taken to be 'the household'.

The majority of the analysis was descriptive in nature and involved summarising findings by region, gender, type of burn, initial first aid and final destination of health service need. The data are presented for the whole cohort, as well as by region. The regional data are important to ensure that the range of local experiences are reflected appropriately: the data for the whole cohort are important in order to provide policymakers with the relevant information for national strategies. Our experiences in the area have highlighted that the local communities feel that they are frequently being asked to participate in surveys - yet feel that nothing changes. So, although it is important to fully understand the baseline situation so that we can evaluate whether our future interventions have been effective, we have also sought ways of presenting our findings to the local community which are easy to understand across languages. The focus has been to highlight the ways in which people access burn care by understanding their pathway from injury to treatment as a first step in co-creating prevention strategies that are locally relevant and implementable.

### Patient and public involvement

The public and former patients were involved in the design of this study through interviews with key personnel, by allowing the researchers to observe their everyday practices (eg, routine cooking and food storage practices), and through community focus groups (n=19). The key stakeholders involved included political leaders, health post workers, household members (including burn survivors), women's groups and schoolteachers. These same groups were involved in the direct dissemination of the community findings using infograms and case-based narratives, and they will help to develop locally relevant and owned awareness campaigns, prevention strategies and educational materials to support appropriate first-aid practices for the next phase of this study.

### RESULTS

Of note, 1305 households were included in the original sampling frame and data were recorded from 1279 (98% of the initial sample). In the 12 months preceding the survey, 2.7% of the households reported at least one serious burn injury (see table 1). Of the three RMs, Laxminiya reported 4 of every 100 households experienced a burn injury; Indrasarowar reported 1:100, while Ama Chhodingmo reported 3:100. More women were

**Table 1**  Percentage prevalence for burn injury in three rural municipalities in Nepal, 2017– 2018 (n=1279 households)

| Percentage responses* | Indrasarowar, Makwanpur (n=415) | Laxminiya, Dhanusha (n=630) | Ama Chhodingmo, Rasuwa (n=234) | Total % | 95% CI |
|---|---|---|---|---|---|
| At least one burn injury | 1.2 | 3.5 | 3.0 | 2.7 | 1.8 to 3.7 |
| Who was burned | | | | | |
| Children | 0.5 | 1.4 | 1.3 | 1.1 | 0.6 to 1.8 |
| Total adults | 0.7 | 2.1 | 1.7 | 1.6 | 1.0 to 2.4 |
| Women | 0.5 | 1.6 | 0.9 | 1.1 | 1.1 to 2.6 |
| Men | 0.2 | 0.5 | 0.9 | 0.5 | 0.5 to 1.6 |
| Cause of burn | | | | | |
| Flames | 0.0 | 1.7 | 0.4 | 0.9 | 0.5 to 1.6 |
| Scalding | 1.0 | 1.6 | 1.7 | 1.4 | 0.8 to 2.2 |
| Other mechanisms | 0.2 | 0.2 | 0.9 | 0.3 | 0.1 to 0.8 |
| Where the burn injury occurred | | | | | |
| At home | 1.2 | 3.5 | 2.1 | 2.5 | 1.7 to 3.5 |
| Away from the home | 0.0 | 0.0 | 0.9 | 0.2 | 0.0 to 0.7 |
| Location of injury | | | | | |
| Head and neck | 0.0 | 0.5 | 0.0 | 0.2 | 0.0 to 0.7 |
| Limbs | 1.0 | 3.0 | 2.1 | 2.2 | 1.5 to 3.1 |
| Trunk | 0.2 | 0.0 | 0.9 | 0.2 | 0.0 to 0.7 |
| Whether the injury occurred during cooking | | | | | |
| While cooking | 1.2 | 3.5 | 2.1 | 2.5 | 1.7 to 3.5 |
| Other activities | 0.0 | 0.0 | 0.9 | 0.2 | 0.0 |

*Figures have been rounded to one decimal place.

affected, scalds were the most frequent cause of injury and the injury was most likely to happen at home; more injuries were reported in adults than children over the 12-month time frame.

Table 2 presents descriptive data on knowledge and practices on burn prevention, first aid and health-seeking behaviour in all 1279 households. Exposure to prevention education was limited, with only 1 in 10 households self-reported exposure to education or information about first-aid treatment for burns. Two-thirds of the households indicated that they believe that most burns were preventable, with a significantly greater proportion (74%) in Laxminiya; interestingly, this is the region where the rates for burn injury were the highest in the previous 12 months.

The most frequent response to first-aid response was the use of water when a family member was scalded (27.4%) and with the application of lotion as the lowest response (5.7%). However, table 2 also outlines regional variations with Laximiniya reporting the highest use of water (38.6%) whereas the highest response in Indrasarowar was to apply tomato, and the highest response in Ama Chhodingmo was to use a range of other options (including mango leaf paste, cow dung, mud and herbs). Households in Ama Chhodingmo also reported that

28.2% said they would not know what to do in such circumstances.

Forty per cent would take off clothing if they caught fire (40.4%); 3 in 10 households reported that they would pour water onto the victim or the victim would jump into water. Only 3 in every 100 households reported that they would stop, drop and roll, while about 1 in 10 reported that they would smother the flames with a cloth. One out of every 10 households indicated that they would not know what to do. These overall figures again hide substantial differences in regional responses (table 2).

The majority of households responded that they would take the family member to hospital following a major burn (73.4%) or to a primary healthcare facility for a minor burn (57.5%). There were regional differences, with only a little over half of households in Ama Chhodingmo reported that they would take the family member to hospital immediately after a large or major burn (55.6%). One-third of all households reported that they would manage minor burn incidents at home immediately after the incident which was consistently reported across the three study sites.

The data relating the experiences of those injured in the previous 12 months have been summarised in an infogram (figure 1) that illustrates patient journeys from

**Table 2** Burn prevention, first aid and health-seeking behaviour in Nepal, 2017–2018

| Percentage responses* | Indrasarowar, Makwanpur (n=415) | Laxminiya, Dhanusha (n=630) | Ama Chhodingmo, Rasuwa (n=234) | Total % | 95% CI |
|---|---|---|---|---|---|
| Burn prevention | | | | | |
| Received education on prevention in the previous 12 months | 13.5 | 10.8 | 6.8 | 10.9 | 9.3 to 12.8 |
| Believe most burns are preventable | 52.3 | 73.8 | 47.4 | 62.0 | 59.3 to 64.7 |
| Believe most people with burns die as a result of their injury | 68.0 | 82.2 | 70.1 | 75.4 | 72.9 to 77.7 |
| Believe that burns can cause bad scars | 85.1 | 96.5 | 85.5 | 90.8 | 89.1 to 92.3 |
| Received first-aid education | | | | | |
| Received education on first aid in the previous 12 months | 15.7 | 10.5 | 5.6 | 11.3 | 9.6 to 13.1 |
| First aid | | | | | |
| Use water | 18.3 | 38.6 | 13.2 | 27.4 | 24.9 to 29.9 |
| Tomato | 39.3 | 5.9 | 14.1 | 18.2 | 16.1 to 20.4 |
| Potato paste | 0.0 | 25.4 | 0.0 | 12.5 | 10.7 to 14.4 |
| Aloe vera | 20.0 | 2.5 | 1.7 | 8.1 | 6.6 to 9.7 |
| Honey | 0.0 | 13.3 | 0.0 | 6.6 | 5.3 to 8.1 |
| Lotion/tube medicine | 5.1 | 4.6 | 9.8 | 5.7 | 4.5 to 7.1 |
| Other (in order of frequency mango leaf paste, cow dung, mud, herbs, various oils and plants) | 5.5 | 5.9 | 32.9 | 10.7 | 9.1 to 12.5 |
| Reported that they would not know what to do | 11.8 | 3.8 | 28.2 | 10.9 | 9.2 to 12.7 |
| Clothing catching on fire | | | | | |
| Stop, drop and roll | 1.7 | 5.9 | 0.0 | 3.4 | 2.5 to 4.6 |
| Smother flames with a cloth | 13.0 | 16.7 | 3.4 | 13.1 | 11.3 to 15 |
| Jump in water | 7.7 | 11.6 | 1.3 | 8.4 | 7 to 10.1 |
| Take off the clothing | 28.4 | 53.8 | 25.6 | 40.4 | 37.7 to 43.2 |
| Pour water | 34.7 | 8.7 | 31.6 | 21.3 | 19.1 to 23.7 |
| Reported that they would not know what to do | 13.0 | 1.7 | 36.3 | 11.7 | 10 to 13.6 |
| Health-seeking behaviour (large or major burn) | | | | | |
| Hospital immediately after incident | 67.7 | 83.8 | 55.6 | 73.4 | 70.9 to 75.8 |
| Primary healthcare facility† immediately after incident | 29.2 | 15.5 | 40.6 | 24.6 | 22.2 to 27.0 |
| Health-seeking behaviour (minor burn) | | | | | |
| Manage at home | 33.7 | 37.6 | 34.6 | 35.8 | 33.2 to 38.5 |
| Primary healthcare facility* | 58.3 | 58.6 | 53.0 | 57.5 | 54.7 to 60.2 |
| Hospital | 1.7 | 2.4 | 5.6 | 2.7 | 1.9 to 3.8 |

*Figures have been rounded to one decimal place.
†Mainly health post but some pharmacy.

initial injury to treatment; each identified case from the data base has been given an icon to reflect whether they are men or women and adult or child. The left-hand side of the icon reflects the type of burn injury, and the right-hand side reflects their RM. For example, case A is a female child from Laximinya, who experienced a flame burn, was treated initially with an alternative form of first aid (tube/lotion) and was transferred by ambulance 5 hours after injury and then she stayed for 35 days in hospital. Ten reported cases did not seek any formal treatment

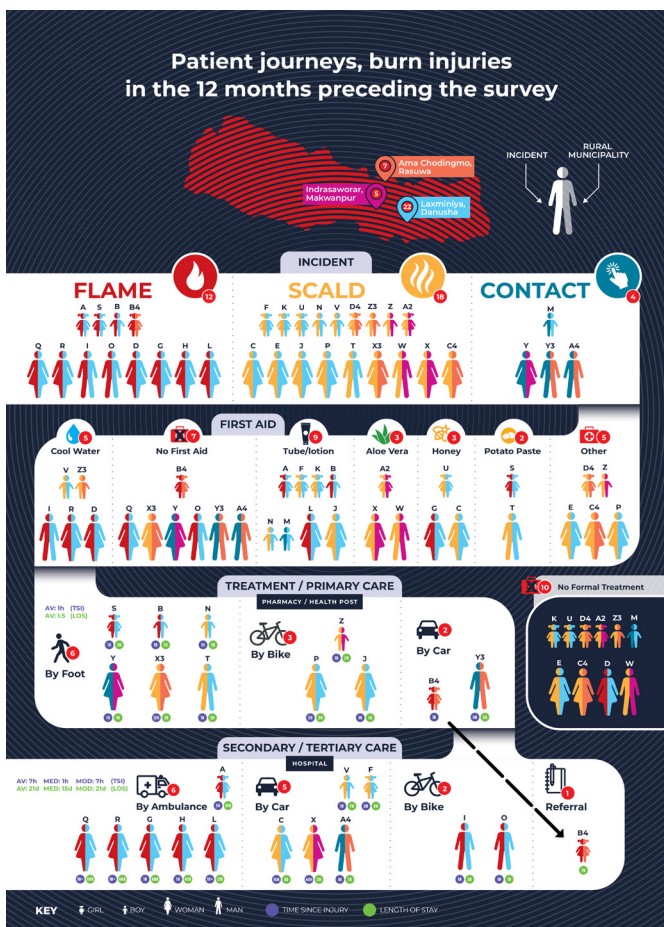

**Figure 1** Patient journeys, burn injuries in the previous 12 months. Each participant has been given a unique identifier and colour coded to show the type of injury and the rural municipality where the participant lived. The average, median and mode are given for (1) time of presentation at a healthcare facility since the injury took place and (2) the length of stay at the location of treatment.

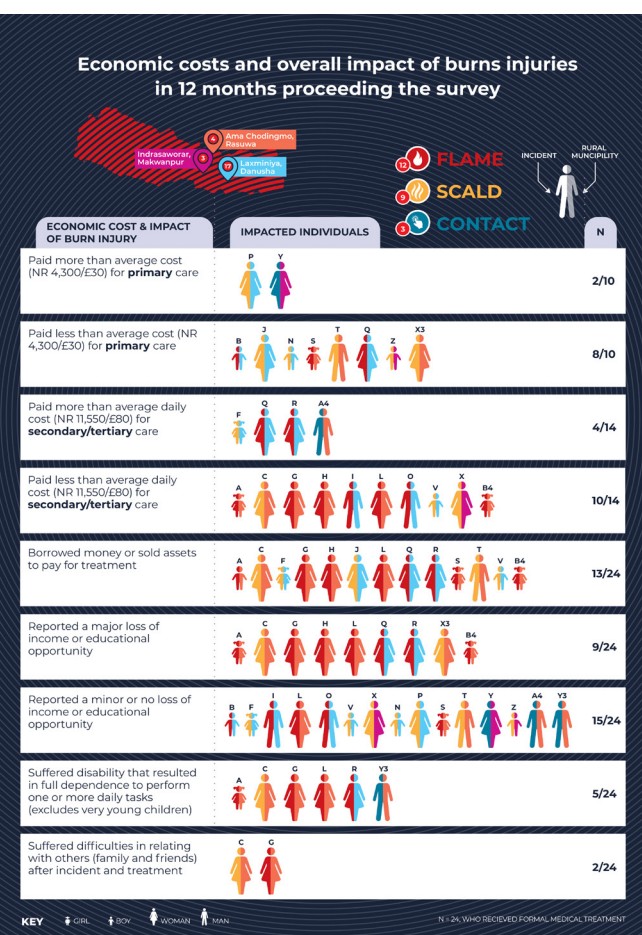

**Figure 2** Economic costs and overall impact of burn injuries in the previous 12 months. Each participant has the same unique identifier as in figure 1.

beyond first aid; 11 cases reported to a primary care facility, one of them (B4) was referred onto secondary/tertiary care where she stayed for 7 days; the remaining 13 cases presented directly to secondary/tertiary care where the average length of hospital stay was 21 days.

Figure 1 shows that only 5 out of 34 burns received appropriate first aid in the form of cooling the wound prior to applying alternative methods for first aid. One in 44 burn victims applied some form of tube-based lotion or medicine as a first-aid measure indicating the availability of some form of 'over-the-counter' burn remedy. Minor burns that can be managed at health posts or pharmacies were filtered appropriately by self-referral, while major or serious burns presented to secondary or tertiary hospitals. The data indicate that most of the serious burn cases requiring admission are disproportionately experienced by women/girls; in all cases, they are linked to cooking activities. Figure 2, which summarises the economic and social costs of 24 individuals who received formal treatment, shows that more than half individuals reported borrowing money or selling assets to cover the costs of

their treatment. One in four suffered a disability that resulted in full dependence to perform one or more daily tasks.

## DISCUSSION

Our data on self-report of serious burn injury, requiring secondary or tertiary care, are substantially higher than we anticipated, compared with the Annual Report,[8] and reflect a substantial number of households who have experienced a serious burn, many of whom needed hospital care and will likely need ongoing management for their pain, disabilities and limitations in function and ability to be self-sufficient. However, responses from Ama Chhodingmo suggested that only a half of households would take serious cases directly to hospital, which may be a reflection of lack of access in terms of resources—financial and otherwise, or physical distance to secondary or tertiary level services.

Together with the background information collected in the initial phases of this study, the survey provides great insight into the risk factors, prevailing knowledge of burn injury at the RM level, as well as the personal financial costs borne as a consequence of burn injuries.

Presenting this information in a simple visual format to local stakeholder groups will enable informed discussion and support the co-creation approach that will empower local communities to implement prevention activities.

Our ability to plot the experiences of individuals from initial injury to length of hospital stay will allow local health teams to work in tandem with local communities to fully explain the importance of initial first aid and the consequences of delayed (or inappropriate) treatment. The use of infograms, to support patient understanding of health messages,[11] has enabled us to summarise the data in an easily digestible format and will allow us to work with a wide range of stakeholders. Although some translation will be required, the emphasis on imagery should support us to disseminate messages to healthcare professionals across the country. Initial work on scoping out a prevention strategy has already highlighted the importance of this local knowledge when interpreting the data and of infograms as a focal point for meaningful discussions when co-creating a context-relevant prevention strategy.

Our data suggest that the key areas for prevention would be cooking within the home setting, particularly for women and children. These data are supported by previous studies included in a systematic review.[7] These authors reported that five out of the eight studies reviewed found more burns in women than men, and the most common location for a burn to occur was inside the house, mostly in the kitchen. One study included in the review reported that 86% of childhood burns occurred in the home. Local knowledge will be extremely important in identifying how this information needs to be built into a programme. First-aid messages need to stress the importance of cool clean water as an initial treatment, when it is available (particularly for scald and flame burns); however, programmes will need to reflect on what is locally available and differentiate between acceptable, alternative cooling options and those that must be avoided (eg, mud). When clothing catches fire, household members need to be aware of the stop, drop and roll method and prioritise that no additional body parts (or other individuals or items) catch fire in the process.

This was a robustly designed study with a strong emphasis on using local knowledge to inform the content of the data collection tool, and substantial emphasis was placed on spending time with the local communities to ensure their full involvement. However, it is important not to over-interpret the findings of the survey at either district or national level; any method that relies on recall is subject to memory bias that can lead to inaccuracies in how the original incident was reported in the survey. In this study, the proportional distribution of households recruited roughly reflected the relative population sizes of the RMs in each district. Only one member of each household, even in households with multiple burn injuries, was interviewed about the incident; the individual selected was the one with the most serious injury meeting the case definition. Given that many households in Nepal are multigenerational and large in size, the person interviewed may not have been aware of all burn incidents within the household and so the possibility exists that on some occasions there may have been a more severe incident within the household which was not captured in this data set. This approach may have resulted in a potential underestimation of burn injuries and any new studies should consider interviewing all those with a serious burn injury.

The team involved in the initial stages of the project are already working, in collaboration with the research team, to co-create appropriate prevention strategies including a wide range of locally relevant activities, where the local community proposed ideas that would make a real difference for their own communities. This is an important initial step in developing locally relevant and owned prevention strategies to help reduce the incidence of burns, ensuring that those with an injury receive the most appropriate first aid and seek appropriate care as quickly as possible.

**Acknowledgements**  We would especially like to acknowledge the support of Mr Bindeshwor Yadav, Chairperson, Laxminiya rural municipality, Dhanusha; Mr Buchung Tamang, Chairperson, Ama Chhodingmo rural municipality, Rasuwa and Mr Jivan Lama, Chairperson, Indrasarowar rural municipality, Makawanpur. Thanks to Arjun Bhattarai for his support in developing the online tool for data capture and support with the sampling frame. The project team would like to express their heartfelt thanks to everyone who took the time to participate in the survey and to the local SAGUN team members who are so enthusiastic to bring about change and improvements in their local communities in Nepal.

**Contributors**  KP: Local lead for SAGUN in Nepal and the local research contact for the survey. EAO: Research assistant with responsibility for assisting with the ethical approval process, organising and analysing the data. RB: Programme manager for InterBurns, supported the logistics for the project and the design of the data collection tool. PEP: Academic lead and responsible for writing the paper. TP: Grant holder and chief investigator. All authors have reviewed the paper prior to submission.

**Funding**  This project is funded by the National Institute for Health Research (NIHR), NIHR Global Health Research Group on Burn Trauma, Grant Reference 16/137/110. The views expressed are those of the authors and not necessarily those of the NIHR or the Department of Health and Social Care.

**Competing interests**  None declared.

**Patient consent for publication**  Not required.

**Ethics approval**  A letter of approval was provided by each of the relevant local governments. The research proposal was reviewed and approved by Nepal Health Research Council (NHRC-Reg. No. 348/2018), with written consent taken from each household.

**Provenance and peer review**  Not commissioned; externally peer reviewed.

**Data availability statement**  Data are available upon reasonable request. Data will be available from NIHR Open Data supplied by NIHR Centre for Business Intelligence, Crown Copyright and database licensed under the Open Govenement Licence (http://www.nationalarchives.gov.uk/doc/open-government-licence/version/3/): Global Health Research (in development). Data will also be archived at Swansea University.

**ORCID iDs**
Edna Adhiambo Ogada http://orcid.org/0000-0003-0181-2505
Patricia E Price http://orcid.org/0000-0003-3767-0131

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
