## [Reviewer comments · BMJ Open]

ARTICLE DETAILS

TITLE (PROVISIONAL)	Burns in Nepal: a participatory, community survey of burn cases and knowledge, attitudes and practices to burn care and prevention in 3 rural municipalities
AUTHORS	Phuyal, Kamal; Ogada, Edna Adhiambo; Bendell, Richard; Price, Patricia Elaine; Potokar, Tom

VERSION 1 – REVIEW

REVIEWER	Dale Wesley Edgar Fiona Stanley Hospital, Western Australia
REVIEW RETURNED	09-Sep-2019

GENERAL COMMENTS	BMJ Open Manuscript -2019-033071 Thank you to the authors for an interesting paper depicting a challenging survey project. The manuscript requires adjustment to be publishable. Please address the following questions: Abstract 1. Please define 'serious burn'.2. The references to study limitations and interpretations should be removed from the Abstract as they detract from the gravity of the overall project from the outset. Let readers determine their judgement on the limitations from the text would be this reviewer's suggestion.3. The Results and Conclusion need to be reconciled – the Abstract should be essentially 'stand alone' in its construction. There isn't any congruent links between these two sections eg nothing in Results points to sharing information and prevention education. Introduction – Please address PDF mark ups + 4. Please include additional references which help to provide context to the relationship of the study results and countries with similarities to Nepal. Methods – Please address PDF mark ups + 5. Is the pooling of data from areas proposed to be diverse and disparate a valid method?6. An indication of the timeframes of the phases of this project would be useful.7. Please define or clarify 'levels of care'. Results – Please address PDF mark ups. Discussion 8. Please explain or clarify how the infographics were incorporated in the survey if at all. There is confusion as to where, when and how they were employed.
---

	The reviewer provided a marked copy with additional comments. Please contact the publisher for full details.
REVIEWER	Saidur Rahman Mashreky Professor, Department of Noncommunicable Diseases, Bangladesh University of Health Sciences (BUHS) Director, Department of Public Health Sciences Centre for Injury Prevention and Research Bangladesh (CIPRB) House B-162; Road 23, New DOHS Mohakhali, Dhaka-1206 Bangladesh
REVIEW RETURNED	26-Sep-2019
GENERAL COMMENTS	Thank you for conducting a valuable study in the field of burn prevention. The objective of the study needs more clarity. It is not clear what are you measuring/exploring in the baseline survey. Methods need little more clarity especially the process of data collection. It is not clear how health-seeking behaviour has been explored. It is in the same survey where prevalence of burn was measured? Needs clarity for what objectives FGD and IDIs were utilized. The conclusion was not appropriately made according to the study result.

VERSION 1 – AUTHOR RESPONSE

Reviewer: 1

Reviewer Name: Dale Wesley Edgar

Institution and Country: Fiona Stanley Hospital, Western Australia

Please state any competing interests or state 'None declared': None Declared

Please leave your comments for the authors below

BMJ Open

Manuscript -2019-033071

Thank you to the authors for an interesting paper depicting a challenging survey project.

The manuscript requires adjustment to be publishable. Please address the following questions:

Abstract

1. Please define 'serious burn'.

This has been completed in the abstract and reiterated in the methods and results.

2. The references to study limitations and interpretations should be removed from the Abstract as they detract from the gravity of the overall project from the outset. Let readers determine their judgement on the limitations from the text would be this reviewer's suggestion.
Completed.

3. The Results and Conclusion need to be reconciled – the Abstract should be essentially 'stand alone' in its construction. There isn't any congruent links between these two sections eg nothing in Results points to sharing information and prevention education.

The Conclusion has been rewritten and now reads:

"The participatory approach used in the study ensured a high response rate. Local partners have been used to help understand the data in context. We have used the results to prepare info-graphics that will now be used to inform the co-creation of locally appropriate prevention strategies".

Introduction – Please address PDF mark ups +
These have been completed on the text.

4. Please include additional references which help to provide context to the relationship of the study results and countries with similarities to Nepal.
References to work in Mongolia and a systematic review on epidemiology studies in Nepal have been completed.

Methods – Please address PDF mark ups +
Completed on the text.

5. Is the pooling of data from areas proposed to be diverse and disparate a valid method?
This is an interesting point. We have clarified why we have done this in the text, to make it clear to readers that it is important to the Ministry of Health in Nepal that both national and regional needs are addressed.

6. An indication of the timeframes of the phases of this project would be useful.
These have been added into the methods: the planning and development work took place in the first 6 months of 2018, with the survey data collected in July-August.

7. Please define or clarify 'levels of care'.
Further information has been added within the results to explain that primary health could include a medical post, a health worker, a female community health volunteer, or a pharmacy

Results – Please address PDF mark ups.
Completed.

Discussion

8. Please explain or clarify how the infographics were incorporated in the survey if at all. There is confusion as to where, when and how they were employed.
We have rewritten the paper to clarify that the infographics were produced as a way to summarise the findings of the survey. We have taken out the different phases of the study as this has caused confusion. Our ongoing work is now using these infographics to create the co-created intervention strategies, which will be addressed in a future paper.

Reviewer: 2

Reviewer Name: Saidur Rahman Mashreky

Institution and Country:

Professor, Department of Noncommunicable Diseases,

Bangladesh University of Health Sciences (BUHS)

Director, Department of Public Health Sciences

Centre for Injury Prevention and Research Bangladesh (CIPRB)

House B-162; Road 23, New DOHS

Mohakhali, Dhaka-1206

Bangladesh

Please state any competing interests or state 'None declared': None declared

Please leave your comments for the authors below

Thank you for conducting a valuable study in the field of burn prevention.

The objective of the study needs more clarity. It is not clear what are you measuring/exploring in the baseline survey.

The objectives have been added at the end of the introduction section.

Methods need little more clarity especially the process of data collection. It is not clear how health-seeking behaviour has been explored. It is in the same survey where prevalence of burn was measured?

Yes this was the same survey; we have removed references to the other phases of the study to clarify.

Needs clarity for what objectives FGD and IDIs were utilized.

The focus group discussions and individual discussions were used to help develop the content of the survey; we hope this is now clear.

The conclusion was not appropriately made according to the study result.

This has been re-written to make them synergistic.

VERSION 2 – REVIEW

REVIEWER	Assoc Prof Dale Edgar Burn Injury Research Node, The University of Notre Dame Australia, Fremantle, Australia.
REVIEW RETURNED	18-Nov-2019

GENERAL COMMENTS	Thank you to the authors for responding to the previous review. The manuscript requires a number of minor adjustments and text edits to be publishable in this reviewer's opinion. Please address the marked up comments added to the revision PDF. The reviewer provided a marked copy with additional comments. Please contact the publisher for full details.
---

REVIEWER	Saidur Rahman Mashreky Professor, Department of Noncommunicable Diseases, Bangladesh University of Health Sciences (BUHS) Director, Department of Public Health Sciences Centre for Injury Prevention and Research Bangladesh (CIPRB) House B-162; Road 23, New DOHS Mohakhali, Dhaka-1206 Bangladesh
REVIEW RETURNED	11-Nov-2019

GENERAL COMMENTS	I am still not clear why two different types of research questions were trying to answer in a single paper. Determining the incidence and exploring the knowledge, attitudes and practices of the community regarding burn care and prevention are two different kind objective. It needs two different kinds of methods is needed for two research questions. Combining these two study makes reader challenging to understand. My suggestion is to make it two different paper.
---

VERSION 2 – AUTHOR RESPONSE

Reviewer: 1

Reviewer Name: Assoc Prof Dale Edgar

Institution and Country: Burn Injury Research Node, The University of Notre Dame Australia, Fremantle, Australia.

Please state any competing interests or state 'None declared': None declared

Please leave your comments for the authors below

Thank you to the authors for responding to the previous review. The manuscript requires a number of minor adjustments and text edits to be publishable in this reviewer's opinion. Please address the marked up comments added to the revision PDF.

Thank you for taking the time to look at the manuscript a second time.

We have gone through the comments on the pdf and amended the text accordingly.

We have expanded the point in the introduction to highlight the difference between hospital reported cases and those from a community setting, together with an additional reference to support the justification for the community study.

You have asked for an alternative for the use of the word 'ward' to help the reader understand how the rural municipalities are determined: we have struggled to find an equivalent so have used the word 'areas' as the material point we are trying to make is about geographical location and lack of access rather than a political/administrative district: we hope this is acceptable in this context.

We have added additional information to the discussion to cover the methodological limitations that you required to be made more explicitly.

You asked for the data in the table to be presented as %; this is the unit of measurement included in the tables so we have amended the label in the table to make this clearer.

We have deleted the sentence in the conclusion about self-referral to the appropriate location for care: although we had toned down this sentence you felt it should not be included.

We hope these changes are satisfactory.

Reviewer: 2

Reviewer Name: Saidur Rahman Mashreky

Institution and Country:

Professor, Department of Noncommunicable Diseases, Bangladesh University of Health Sciences (BUHS)

Director, Department of Public Health Sciences

Centre for Injury Prevention and Research Bangladesh (CIPRB)

House B-162; Road 23, New DOHS

Mohakhali, Dhaka-1206

Bangladesh

Please state any competing interests or state 'None declared': None declared

Please leave your comments for the authors below

I am still not clear why two different types of research questions were trying to answer in a single paper.

Determining the incidence and exploring the knowledge, attitudes and practices of the community regarding burn care and prevention are two different kind objective. It needs two different kinds of methods is needed for two research questions. Combining these two study makes reader challenging to understand.

My suggestion is to make it two different paper.

Thank you for this suggestion. We debated at length at the initial stages of writing this paper as to whether or not this should be two papers or one. The authors are of the opinion that this fits much better as a single paper. The data were all collected as part of one community survey which included separate sections related to the different objectives. It is really important for us to share the data with the local community in a way that links the epidemiological data with the direct experiences of each of the individuals in the three rural municipalities, so that we can build information around the patient experience in terms of the initial response from family and friend for first aid, through to the presentation at the definitive location for burn care. By linking these two datasets we can track the

time taken to get to definitive care, demonstrate the mode of transport, and length of hospital stay for each of the cases reported as a 'burn case' in the previous 12 months. This is one of the original aspects of the study and as such we would like to keep the paper in its current format. Thank you for taking the time to consider this manuscript.

VERSION 3 – REVIEW

REVIEWER	Dale Edgar Burn Injury Research Node, The University of Notre Dame Australia
REVIEW RETURNED	05-Jan-2020

GENERAL COMMENTS	Thank you to the authors for responding to all but one comment from the previous review. It is the opinion of this reviewer that the manuscript is worth of publication with minor adjustments - please see attached mark ups. The reviewer provided a marked copy with additional comments. Please contact the publisher for full details.
---

VERSION 3 – AUTHOR RESPONSE

Thank you for the comments from the reviewer. We have accepted the suggested changes to wording in the introduction by including the sentence "Understanding the true morbidity and mortality impacts of burn injury in Nepal is challenging" on page 3, and replaced 'bring about' with "proactively develop" (page 4). We have also added a clause on page 4 to confirm that the survey also had the purpose of establishing a baseline against which future studies could be compared; we had thought this was self-evident, but are happy to add it for the purposes of clarity.

In Table 2, the reviewer suggests removing a line from the Table and adding it to the text, as the statistic does not match the rest of the sub-section within the table. We have discussed this in detail: we have created an additional sub-section within the table so that this item stands separated from the rest of the items on First Aid. We hope this simplifies the flow of the information within Table 2.

The reviewer requested that we consider the best way to present an item in Table 2, and proposed that we label the category as 'no knowledge of options'. We have discussed this in detail and feel this changes the nature of the sentiment captured in the survey; we have reworded the item to read 'reported that they would not know what to do'. We hope this captures the data clearly.

We have accepted the minor editorial requests from the reviewer as follows:

1. we have deleted the semi-colon and replaced it with a full stop (page 10), changing associated punctuation to fit the new sentence structure
2. we have replaced numerals with words in all places requested (page 10 and Tables 1 and 2).

We hope that this set of changes will allow the paper to go forward for publication. With many thanks from the research team and all the participants in the study.